# The predictive role of circulating telomerase and vitamin D for long-term survival in patients undergoing coronary artery bypass grafting surgery (CABG)

Mahtab Zarei[1], Mahdi Najafi[2,3]*, Elnaz Movahedi[1], Mohamad Hassan Javanbakht[1], Yun-Hee Choi[4], Mehdi Yaseri[5], Arash Shirvani[6], Frank W. Sellke[7], Saverio Stranges[4,8,9]

1 Department of Cellular and Molecular Nutrition, Faculty of Nutritional Sciences & Dietetics, Tehran University of Medical Sciences, Tehran, Iran, 2 Department of Medical Biophysics, Schulich School of Medicine & Dentistry, Western University, London, Ontario, Canada, 3 Department of Anesthesiology, Tehran Heart Center, Tehran University of Medical Sciences, Tehran, Iran, 4 Department of Epidemiology and Biostatistics, Schulich School of Medicine & Dentistry, Western University, London, Ontario, Canada, 5 Department of Epidemiology and Biostatistics, School of Public Health, Tehran University of Medical Sciences, Tehran, Iran, 6 Department of Medicine, Section of Endocrinology, Nutrition, and Diabetes, Vitamin D, Skin and Bone Research Laboratory, Boston University Medical Campus, Boston, Massachusetts, United States of America, 7 Division of Cardiothoracic Surgery, Alpert Medical School of Brown University, Rhode Island Hospital, Providence, Rhode Island, United States of America, 8 Department of Family Medicine, Schulich School of Medicine & Dentistry, Western University, London, Ontario, Canada, 9 Department of Population Health, Luxembourg Institute of Health, Strassen, Luxembourg

* mahdi.najafi@uwo.ca, najafik@sina.tums.ac.ir

**Data Availability Statement:** All relevant data are within the paper and its Supporting Information files.

## Abstract

### Backgrounds

Cardiovascular disease (CVD) is the leading cause of mortality all over the globe. Inflammation is believed to play a pivotal role in the pathophysiology of CVD. While there are studies on the interrelationship of telomerase and vitamin D and their involvement in CVD, their independent contributions to long-term outcomes in patients with CVD are not well-defined. This study aimed to investigate the association of both telomerase and vitamin D concentrations with 10-year survival among candidates of coronary artery bypass grafting (CABG) surgery.

### Methods

Participants were 404 patients from Tehran Heart Center-Coronary Outcome Measurement (THC-COM) cohort who were recruited from CABG surgery candidates in 2006. In addition to demographic and clinical data including risk factors for coronary artery disease, laboratory parameters such as markers of inflammation as well as baseline serum 25-hydroxy vitamin D [25(OH)D] and telomerase concentrations were measured. Cardiac function indexes alongside outcome measures such as mortality and survival days were recorded for every patient up to 10 years after CABG. Cox-proportional hazard model was used to study the association between all-cause mortality and research parameters.

**Funding:** MHJ was awarded grant number 33885 from Research Undersecretary of Tehran University of Medical Sciences. http://medicine. tums.ac.ir/research/ The funders had no role in study design, data collection and analysis, decision to publish, or preparation of the manuscript.

**Competing interests:** The authors have declared that no competing interests exist.

**Abbreviations:** 25(OH)D, 25-hydroxy vitamin D; BMI, Body mass index; CABG, Coronary artery bypass grafting; CAD, Coronary artery disease; Cr, Serum creatinine; CRP, C-reactive protein; CVA, Cerebrovascular accident; CVD, Cardiovascular disease; EF, Ejection fraction; MI, Myocardial infarction; NYHA, New York Heart Association; PVD, Peripheral vascular disease.

## Results

The mean serum telomerase enzyme level was 24.92 ±21.4 nmol/L and the mean serum 25 (OH)D was 27.27±10.3 ng/mL. 10-year mortality was reported in 64 (15.8%) patients. 25 (OH)D was categorized into three groups (<20, 20–30, and >30) and the cut-point for telomerase was set at 25.0 nmol/L. In Cox regression analysis, higher levels of telomerase (>25 nmol/L) were significantly associated with longer survival (p = 0.041), whereas 25(OH)D concentrations were not associated with survival time. Further analysis showed that telomerase concentrations significantly predicted survival only in the presence of insufficient levels of 25(OH)D (20–30 ng/mL) (p = 0.037).

## Conclusions

Telomerase can be regarded as a potential predictor of long-term outcomes in patients who underwent CABG. However, the association of telomerase with the mortality may be modified by vitamin D concentrations.

## Introduction

Cardiovascular disease (CVD) is the leading cause of death worldwide[1]. The burden of CVD is increasingly higher in low- and middle-income countries which are undergoing a rapid epidemiological transition, with a projected increase in CVD mortality from 17.3 million to 23.6 million in 2030[2]. CVD is currently the first cause of mortality in Iran and the prevalence is predicted to double in 2025 as compared to two decades earlier[3].

In addition to some known risk factors such as hypertension, obesity, smoking, diabetes, low physical activity, hyperlipidemia, and unhealthy diet, there are many other proposed risk markers that have been shown to be associated with CVD. The potential relevance of some of these biomarkers such as telomerase and vitamin D may be attributed to treating CVD as an inflammatory process[4–8].

Telomeres are structures made up of DNA and proteins. They are found at the end of linear chromosomes in eukaryotes and protect genome from degradation and interchromosomal fusion. Telomeres represent the biological age of the cell and shorten with each cell division. Gradual shortening of telomere makes cells more vulnerable to chronic diseases such as CVD and may lead to more rapid and progressive senescence[9]. The association of telomere length with the incidence of CVD and its risk factors has been reported in epidemiological studies [5,10–13]. The same is found in atherosclerotic plaques and is associated with plaque instability in coronary artery disease (CAD) as a result of two major mechanisms underlying CVD, inflammation and oxidative stress[5,14]. However, the relationship between telomere length and CVD mortality is still debated[9,10,15]. The paucity of knowledge about telomerase is more significant.

Telomerase enzyme' activity results in telomere extension. However, it is normally present in germline and hematopoietic cells and the expression of this enzyme does not usually occur in somatic cells. Therefore, somatic cells undergo progressive shortening by aging[16]. While telomerase is recognized for its involvement in cancer cell immortality and for monitoring anti-cancer therapy, its role in CVD prevention and progression is not clearly defined. There is however promising evidence that in spite of the low level of telomerase expression in mammalian heart, its function is significant. An interesting experimental study showed that it

increases to as much as 4.5 fold in response to injury despite its decreasing trend by aging[17]. Studies on the relationship between telomerase and long-term outcomes in CVD patients are lacking.

There is compelling evidence that low levels of vitamin D are associated with CVD[18–20] even though the mechanism of this relationship has yet to be defined. However, reports on vitamin D relationship with long term CVD outcomes are not conclusive[21–23]. Also, there is still a big controversy on the potential role of vitamin D supplementation in CVD prevention[20,21,24].

Studies have shown the relationship of vitamin D level with telomere length[25]. Some studies have attributed this to the anti-inflammatory role of vitamin D reflected by the inverse relationship between the level of vitamin D and the proliferation of lymphocytes, CRP, gamma interferon and interleukin II. Others have found a direct effect of vitamin D on telomeres[25–27]. There are few reports that vitamin D may control the activity of telomerase and provide cellular protection[4]. The interaction of vitamin D and telomerase in patients with CVD and their effect on long-term outcome has not been studied. This study aimed to assess the relationship of Vitamin D with telomerase enzyme and their association with 10-year survival of patients who underwent isolated coronary artery bypass graft (CABG) in a cardiac tertiary center.

## Materials & methods

This study was carried out on a group of patients who participated in Tehran Heart Center Coronary Outcome Measurement (THC-COM) cohort study. Patients were recruited over a six months period (April 2006-September 2006) among candidates for isolated coronary artery bypass graft (CABG) surgery, as previously described[28–30]. Patients underwent CABG because they had at least 50% arterial obstruction based on their angiography in one or more coronary vessels. Patients who needed valvular surgery and/ or ventricular aneurism repair as well as patients on dialysis were excluded. The protocol was approved by Tehran Heart Center/ Tehran University of Medical Science review board under number 0507150 and conformed to the Helsinki Declaration. Participants signed provided written informed consent after explanation of the main objective of the study. Demographic and anthropometric data such as sex, height, weight, education, body mass index (BMI) as well as known risk factors including smoking alongside other addictions, diabetes, blood pressure, lipid profile, and inflammatory biomarkers were collected in a dedicated database. Systolic and diastolic blood pressure was measured by a digital arterial pressure gauge in sitting position after 5 minutes of rest. Hypertension was defined as systolic pressure >140 mmHg and/or diastolic pressure >90 mmHg in at least two separate measurements or if the individual was taking antihypertensive medications. Diabetes mellitus was defined as previous diagnosis, specific treatment administration (oral drug or insulin), fasting blood sugar at least 126 mg/dL or HbA1c > 6.5%. Clinical assessment and pre-operative cardiac status was measured by several variables such as: number of diseased vessels, New York Heart Association (NYHA) functional class, left ventricular ejection fraction and the European system for cardiac operative risk evaluation (EuroSCORE).

### Biochemical measurements

Blood samples were taken on the day of operation after 14 hours of fasting. At 4 ˚C, centrifugation was performed at 3500 rpm for 10 minutes to separate the serum. All laboratory analyses performed on blood samples to report prescheduled laboratory data including serum total cholesterol, HDL-C, LDL-C, serum TG, fasting blood glucose, and HbA1C[29–31]. Then, the serum isolated from 10 mL of the blood sample of each patient was transferred to the sterile

micro tubes. Micro tubes contents stored frozen at -80 ˚C. Baseline serum 25-hydroxy vitamin D [25(OH)D] assessment was performed by chemiluminescence method through ELISA Vitamin D assay (DIAsource ImmunoAssays SA). 25(OH)D was measured and reported in ng/mL in this study. Baseline serum Telomerase enzyme level was also measured using ELISA (Bioassay Technology Laboratory) and reported in nmol/L.

## Outcome measurement (recording survival data and mortality assessment)

Patients' follow-up schedule has been discussed in detail previously[30]. All patients were followed up to 10 years. 10-year survival was assessed through telephone interviews. For non-responders, mortality tracking as well as the exact time and etiology of death was noted through telephone interviews with patients' relatives or friends. For the rest of patients data were obtained by checking up the registration of deaths in Tehran Heart Center archive, through correspondence with National Organization for Civil Registration, or through registry system of large Tehran cemetery. For non-responders, the last attendance at THC for receiving any services was considered as the last time of follow-up.

## Statistical analysis

To analyze vitamin D and telomerase on a reasonable number of patients out of all participants in THC-COM, the sample size was calculated to achieve 80% power with type I error of 0.05 to detect a hazard ratio as large as 2 between any two groups, with a minimum number of 68 events needed. In addition, we assumed a median survival time of 6 years, censoring rate of 80% and planned follow up time of 10 years. The sample size as large as 411 subjects was required to observe 68 events from the sample[32].

   We checked the normality assumption for the quantitative variables with Shapiro-Wilks test and Q-Q plot. Continuous variables were presented as mean and standard deviation (SD). Categorical variables were expressed as frequency and percentage. To evaluate the relationship between different factors and death at 10 years we used t-test, Mann-Whitney test and Chi-square test, whenever appropriate. To obtain the optimum cutoff value for categorizing the telomerase we used decision tree analysis. It showed that two categories with one cut-off value might be the best model. We calculated cut-off point with ROC curve. 25(OH)D was categorized into three groups (<20, 20–30, and >30) based on previous reports in literature[25,33–35]. We estimated the survival curves for each group using the Kaplan-Meir method and performed a log-rank test for comparing survival curves of different groups. In addition, we used Cox-proportional hazard models to assess the bivariate association of each variable with the patients' survival time. Variables with a P-value less than 0.2 in the bivariate analysis were added into a multivariable Cox regression model to obtain adjusted hazard ratios. We analyzed the interaction between vitamin D and telomerase in multivariate model as well. A P-value of $< 0.05$ was considered statistically significant. Softwares IBM SPSS statistics for windows version 22 (Armonk, NY: IBM Corp.) and R (R Core Team, Vienna, Austria: R foundation for Statistical Computing, 2019) were used to conduct the analyses.

## Results

In this study, 404 patients 110 women (27.2%) and 294 men (72.8%) were enrolled. The mean serum telomerase enzyme level was 24.92 ±21.4 nmol/L and the mean serum 25(OH)D was 27.27±10.3 ng/mL. The cut-point for telomerase was set on 25 based on the statistical method mentioned earlier and 25(OH)D was categorized into three groups (<20, 20–30, and >30). The log rank test indicated that higher level of telomerase (>25 nmol/L) was significantly associated with longer survival time (P = 0.041). Further analysis revealed a significant interaction

between vitamin D and telomerase in relation to survival. It showed that telomerase was significantly associated with the survival only in the presence of insufficient levels of 25(OH)D (20–30 ng/mL) (p = 0.037).

Overall, 10-year mortality was reported in 64 (15.8%) patients. There were only three patients lost to follow-up and their survival data were missing. Median (25th-75th interquartile range) follow-up time for survival was 79.8 months (75.5–81.4). With regards to target variables, telomerase and 25(OH)D were measured in 402 and 399 patients respectively. Missing was due to technical problems in handling low volume samples and they were not substituted with the other samples as we did not find any patient-related issue. Demographic and clinical characteristics, as well as laboratory data and measures of cardiac function are summarized in Table 1. The mean age of CAD patients at the time of recruitment was 59.54±8.85. Outcome measure was considered 10-year mortality in Table 1. The same variables were analyzed using Cox regression to report hazard ratio for mortality considering survival data as outcome measure (Table 2).

Fig 1 shows the relationship between telomerase level and survival times in CAD patients who underwent CABG. As depicted in Fig 2, the mean survival rate for CAD patients with 25(OH)D <20 was 73.6 months, for CAD patients with 25(OH)D between 20–30 was 72.6 months and for those with 25(OH)D >30 was 73.9 months. The log rank test indicated that there was no statistically significant difference in the survival rate of patients with three different levels of 25(OH)D (P = 0.284). Furthermore, we found that telomerase level >25 was seen in 18.1% of patients who had MI and 25.9% of patients who had no history of MI (OR = 0.66 95% CI = (0.41–1.06), p = 0.08).

Then we performed multivariate Cox-regression using variables with p values less than 0.2 from bivariate analyses to develop an adjusted model for predictors of 10-year survival. We did not include EuroSCORE because it contains the same variables included in our analysis. MI was not significantly associated with mortality in multivariate model. We did not include MI in final model whatsoever due to a possible interaction between MI and telomerase. Age, opium addiction, history of CVA, peripheral vascular disease (PVD), serum creatinine, CRP, and cardiac ejection fraction (EF) were significantly associated with mortality as shown in Table 3. Specifically, for each 10-year increase in age, the risk of mortality doubled.

## Discussion

This study showed that vitamin D was not significantly associated with survival time alone, while telomerase level was an independent predictor of long-term mortality. However, there was an interaction between vitamin D and telomerase with regards to their relationship with the survival; only at insufficient levels of 25(OH)D telomerase was associated with survival times. To the best of our knowledge, this is the first study investigating the relationship between serum 25(OH)D levels, serum telomerase enzyme and 10-year survival in CVD patients who underwent CABG.

It is well-known that telomerase level is a determinant of telomere length. So, it can serve as an earlier predictor of cellular longevity than telomere[36]. Moreover, it is evident from the literature that telomerase has more unique features in cardiac patients that are independent from its ability to repair and elongate the telomere. For instance it offers an ability for cell protection when telomeres become shorter[37]. Additionally, an interesting experimental study showed that reactivation of telomerase in aged mice can regenerate tissue degeneration[38]. Accordingly, another experimental study in neonatal, adult, and cryo-injured transgenic mice showed that while telomerase trend is decreasing in process of aging, it can increase 4.5 fold in

**Table 1. Demographic, clinical and preclinical parameters at the beginning and their relationship with 10-year mortality.**

| Variable | Categories | Total | Death | | P-value |
|---|---|---|---|---|---|
| | | | no | yes | |
| *Demographic parameters* | | | | | |
| Gender | Women | 110 (27.6) | 97 (29) | 13 (20.3) | 0.156 * |
| | Men | 289 (72.4) | 238 (71) | 51 (79.7) | |
| Age (year) | | 59.5 (8.9) | 58.8 (8.6) | 64.1 (8.8) | **0.000** † |
| Education | Illiterate/ Basic literacy | 225 (56.4) | 180 (53.7) | 45 (70.3) | **0.021** ‡ |
| | Elementary & high school | 114 (28.6) | 102 (30.4) | 12 (18.8) | |
| | Higher education | 60 (15) | 53 (15.8) | 7 (10.9) | |
| *Clinical parameters* | | | | | |
| BMI (kg/m$^2$) | | 27.4 (4.0) | 27.40 (4.0) | 27.5 (4.4) | 0.906 † |
| Hypertension | | 196 (49.1) | 160 (47.8) | 36 (56.3) | 0.213 * |
| Hyperlipidemia | | 288 (72.2) | 246 (73.4) | 42 (65.6) | 0.202 * |
| Family history | | 190 (47.6) | 162 (48.4) | 28 (43.8) | 0.499 * |
| Diabetes | | 179 (44.9) | 144 (43.0) | 35 (54.7) | 0.085 * |
| PVD | | 106 (26.6%) | 79 (23.6%) | 27 (42.2%) | 0.002 * |
| History of Myocardial infarction | | 198 (49.9) | 160 (47.9) | 38 (60.3) | 0.071 * |
| History of CVA | | 16 (4) | 11 (3.3) | 5 (7.8) | 0.091 * |
| Smoking | | 138 (34.6) | 111 (33.1) | 27 (42.2) | 0.163 * |
| Opium addiction | | 59 (14.8) | 43 (12.8) | 16 (25.0) | **0.012** * |
| Alcohol | | 47 (12.6) | 39 (12.5) | 8 (13.1) | 0.902 * |
| *Determinants of cardiac Situation* | | | | | |
| NYHA function class | 1 | 134 (33.6) | 107 (31.9) | 27 (42.2) | 0.509 ‡ |
| | 2 | 202 (50.6) | 176 (52.5) | 26 (40.6) | |
| | 3 | 63 (15.8) | 52 (15.5) | 11 (17.2) | |
| EF | | 48.6 (10.1) | 49.5 (9.7) | 44.1 (11.1) | **0.000** † |
| Number of diseased vessels | 1 | 15 (3.8) | 12 (3.6) | 3 (4.7) | 0.774 ‡ |
| | 2 | 79 (19.8) | 68 (20.3%) | 11 (17.2) | |
| | 3 | 305 (76.4) | 255 (76.1%) | 50 (78.1) | |
| EuroSCORE | | 2.45 (2.2) | 2.25 (2.0) | 3.64 (2.8) | **0.000** † |
| *Paraclinical parameters* | | | | | |
| BUN | | 40.5 (12.4) | 39.6 (11.3) | 45.8 (16.3) | **0.000** † |
| Creatinine | | 1.3 (0.3) | 1.3 (0.25) | 1.4 (0.35) | **0.000** † |
| Hematocrit | | 42.1 (4.3) | 42.0 (4.3) | 43. (4.1) | 0.596 † |
| Cholesterol | | 161.6 (46.0) | 161 (46.6) | 165 (44.9) | 0.525 † |
| LDL | | 87.4 (40.1) | 87.4 (41.3) | 88.1 (35.3) | 0.898 † |
| HDL | | 40.4 (8.9) | 40.5 (8.6) | 40.1 (10.1) | 0.745 † |
| TG | | 176.4 (92.3) | 174.8 (86.4) | 183.5 (119.7) | 0.489 † |
| Hb A1c | | 6.2 (1.8) | 6.1 (1.7) | 6.3 (2.0) | 0.633 † |
| Albumin | | 4.7 (0.3) | 4.7 (0.3) | 4.6 (0.3) | 0.163 † |
| CRP | | 6.7 (5.5) | 6.5 (5.5) | 8.0 (5.6) | 0.052 † |
| LP | | 32.5 (26.7) | 32.3 (27.0) | 33.3 (25.2) | 0.793 † |
| 25(OH)D | ≤20 | 105 (26.6) | 85 (25.7) | 20 (31.7) | 0.305† |
| | 20–30 | 132 (33.5) | 116 (35.0) | 16 (25.4) | |
| | >30 | 157 (39.8) | 130 (39.3) | 27 (42.9) | |
| Telomerase enzyme | ≤25 | 310 (78.1) | 253 (76) | 57 (89.1) | **0.020** * |
| | >25 | 87 (21.9) | 80 (24) | 7 (10.9) | |

(*Continued*)

**Table 1.** (Continued)

| Variable | Categories | Total | Death | | P-value |
| --- | --- | --- | --- | --- | --- |
| | | | no | yes | |
| Survival (months) | | 73.4 (17.7) | 76.9 (11.9) | 54.7 (28.8) | **0.000** † |

BMI, body mass index; EF, ejection fraction; CVA, cerebrovascular accident; NYHA, New York Heart Association; CRP, C-reactive protein, LP, alpha lipoprotein; LDL, low density lipoprotein cholesterol; HDL, high density lipoprotein cholesterol; TG, triglyceride; BUN, blood urea nitrogen; PVD, peripheral vascular disease; 25(OH)D, 25-hydroxy vitamin D

† Reported in mean (SD), t-test

‡ Reported in number (percentage), Mann-Whitney test.

* Reported in number (percentage), Chi-Square test.

response to injury[17]. This is promising especially concerning cardiomyocyte damage following myocardial infarction (MI) or heart failure in CVD patients.

Inflammation and oxidative stress not only contribute to CVD pathophysiology, they are responsible for enhancing telomere shortening and genomic instability too. This hypothesis delineates the interrelationship of vitamin D and telomeres in CVD etiology and the disease progress[25,26] and is reflected in our study by prominent role of CRP as a measure of inflammation. Also, it may show that relatively higher levels of 25(OH)D provides enough protection against inflammation and oxidative stress at cellular level to improve long-term outcome. With lower levels of 25(OH)D, the cell is more vulnerable and probably needs higher level of telomerase for survival. The relationships between telomeres and CVD and between telomeres and vitamin D have been discussed in the literature. In a systematic review and meta-analysis in 2014, 24 studies and 43725 participants were included. Among them, 8400 patients had CVD, 5566 with CAD and 2834 with cerebrovascular involvement. The authors concluded that there was a negative relationship between telomere length and risk of CVD[39]. A positive relationship between higher levels of vitamin D and the length of telomere was reported in a cohort study involving 2160 women aged 18 to[26]. While there are more such reports about telomeres[25,27], similar investigations on telomerase and vitamin D relationship are lacking. Just partially relevant, a double-blind randomized clinical trial was conducted on a small group of 37 overweight African-American adults aged 19 to 50 to examine the effect of 16 weeks supplementation with vitamin D on telomerase activity in peripheral blood mononuclear cells (PBMCs). In that trial 19 subjects were randomly assigned to the vitamin D group and 18 subjects to the placebo group. Findings confirmed a significant increase in the activity of telomerase (an increase of 19.2% in PBMCs) in those who took supplemental vitamin D[4]. While the study did not carry out in CAD patients and was not about the outcome, it showed that the activity and the level of telomerase was probably correlated with the level of vitamin D. To add, the relationship between vitamin D level and CVD outcome is non-linear[10]. In a cohort study in 2016, the relationship between 25(OH)D levels and long-term mortality in patients with acute MI was nonlinear and U-shaped. For those who had 25(OH)D level of <10 ng/mL and >30 ng/mL, the estimated hazard rate was 3.02 times higher compared to those with level 20–30 ng/mL[34]. We employed similar categorization which may explain why telomerase prediction of long-term survival mediated by 25(OH)D levels in our study. At the present time, it is unclear if other factors may predict outcomes at lower levels of 25(OH)D.

Survival rates as determinants of outcome were not significantly different among three different levels of serum 25(OH)D in current study. There is no consensus on the impact of vitamin D on CVD outcome. A review of cohort studies in 2019 revealed that there is a large inconsistency between the studies on vitamin D with regards to the relationship with the CVD

**Table 2. Demographic, clinical and preclinical parameters at the beginning in patients undergoing coronary artery bypass grafting surgery and their relationship with 10-year survival.**

| Variable | Categories | HR | 95% confidence interval | | P-value |
|---|---|---|---|---|---|
| | | | Lower | Upper | |
| *Demographic parameters* | | | | | |
| Gender | Woman | Ref | | | 0.153 |
| | Man | 1.559 | 0.848 | 2.868 | |
| Age (year) | | 1.071 | 1.039 | 1.104 | **0.000** |
| Education | Illiterate/ Basic literacy | Ref | | | |
| | Elementary & high school | 0.503 | 0.266 | 0.952 | **0.035** |
| | Higher education | 0.578 | 0.261 | 1.283 | 0.178 |
| *Clinical parameters* | | | | | |
| BMI (kg/m$^2$) | | | | | |
| Hypertension | | 1.364 | 0.832 | 2.235 | 0.218 |
| Hyperlipidemia | | 0.719 | 0.429 | 1.204 | 0.210 |
| Family history | | 0.840 | 0.512 | 1.376 | 0.488 |
| Diabetes | | 1.554 | 0.950 | 2.543 | 0.079 |
| PVD | | 2.247 | 1.368 | 3.692 | **0.001** |
| History of Myocardial infarction | | 1.556 | 0.939 | 2.579 | 0.086 |
| History of CVA | | 2.457 | 0.986 | 6.123 | 0.054 |
| Smoking | | 1.392 | 0.848 | 2.287 | 0.191 |
| Opium addiction | | 2.025 | 1.150 | 3.567 | **0.015** |
| Alcohol | | 1.030 | 0.490 | 2.167 | 0.937 |
| *Determinants of cardiac Situation* | | | | | |
| NYHA function class | I | Ref | | | |
| | II | 0.601 | 0.351 | 1.031 | 0.064 |
| | III | 0.863 | 0.428 | 1.739 | 0.679 |
| EF | | 0.955 | 0.932 | 0.977 | **0.000** |
| Number of diseased vessels | 1 | Ref | | | |
| | 2 | 0.634 | 0.177 | 2.237 | 0.484 |
| | 3 | 0.741 | 0.231 | 2.377 | 0.614 |
| EuroSCORE | | 1.164 | 1.067 | 1.269 | **0.001** |
| *Paraclinical parameters* | | | | | |
| BUN | | 1.033 | 1.016 | 1.049 | **0.000** |
| Creatinine | | 3.647 | 2.010 | 6.618 | **0.000** |
| Hematocrit | | 1.014 | 0.957 | 1.074 | 0.645 |
| Cholesterol | | 1.001 | 0.996 | 1.006 | 0.590 |
| LDL | | 1.000 | 0.994 | 1.006 | 0.912 |
| HDL | | 0.993 | 0.965 | 1.022 | 0.625 |
| TG | | 1.001 | 0.998 | 1.003 | 0.573 |
| Hb A1c | | 1.026 | 0.900 | 1.168 | 0.703 |
| Albumin | | 0.537 | 0.249 | 1.161 | 0.114 |
| CRP | | 1.027 | 1.002 | 1.052 | **0.031** |
| LP | | 1.001 | 0.992 | 1.011 | 0.749 |
| 25(OH)D | 20–30 | Ref | | | |
| | ≤20 | 1.639 | 0.848 | 3.167 | 0.142 |
| | >30 | 1.519 | 0.817 | 2.823 | 0.186 |

(*Continued*)

**Table 2.** (Continued)

| Variable | Categories | HR | 95% confidence interval | | P-value |
|---|---|---|---|---|---|
| | | | Lower | Upper | |
| Telomerase enzyme | ≤25 | Ref | | | |
| | >25 | 0.471 | 0.224 | 0.987 | **0.046** |

BMI, body mass index; EF, ejection fraction; CVA, cerebrovascular accident; NYHA, New York Heart Association; CRP, C-reactive protein, LP, alpha lipoprotein; LDL, low density lipoprotein cholesterol; HDL, high density lipoprotein cholesterol; TG, triglyceride; BUN, blood urea nitrogen; PVD, peripheral vascular disease; 25(OH)D, 25-hydroxy vitamin D

outcome. Out of 12 studies on cardiovascular and coronary outcome that was assessed, the vitamin D showed negative association in nine and no association in three studies[40]. Moreover, we showed previously in the same population and setting that the level of 25(OH)D was significantly lower in CAD patients compared to control group[7,8,41]. Also, the serum level of 25(OH)D was not significantly correlated with the number of diseased vessels as a representative of the extent of cardiac involvement [7,8] which is in accordance with our findings in this study. It is noteworthy that serum levels of 25(OH)D for almost all of our patients (98.5%) was lower than recommended ranges for general population and especially for cardiovascular patients[42]. So, our patients all were indeed in the same category. There are other possible explanations for controversial relationship of vitamin D and cardiovascular outcome. Patients who receive vitamin D for supplementation usually are prescribed a complex treatment containing both vitamin D and supplemental calcium while the impact of these two on CVD

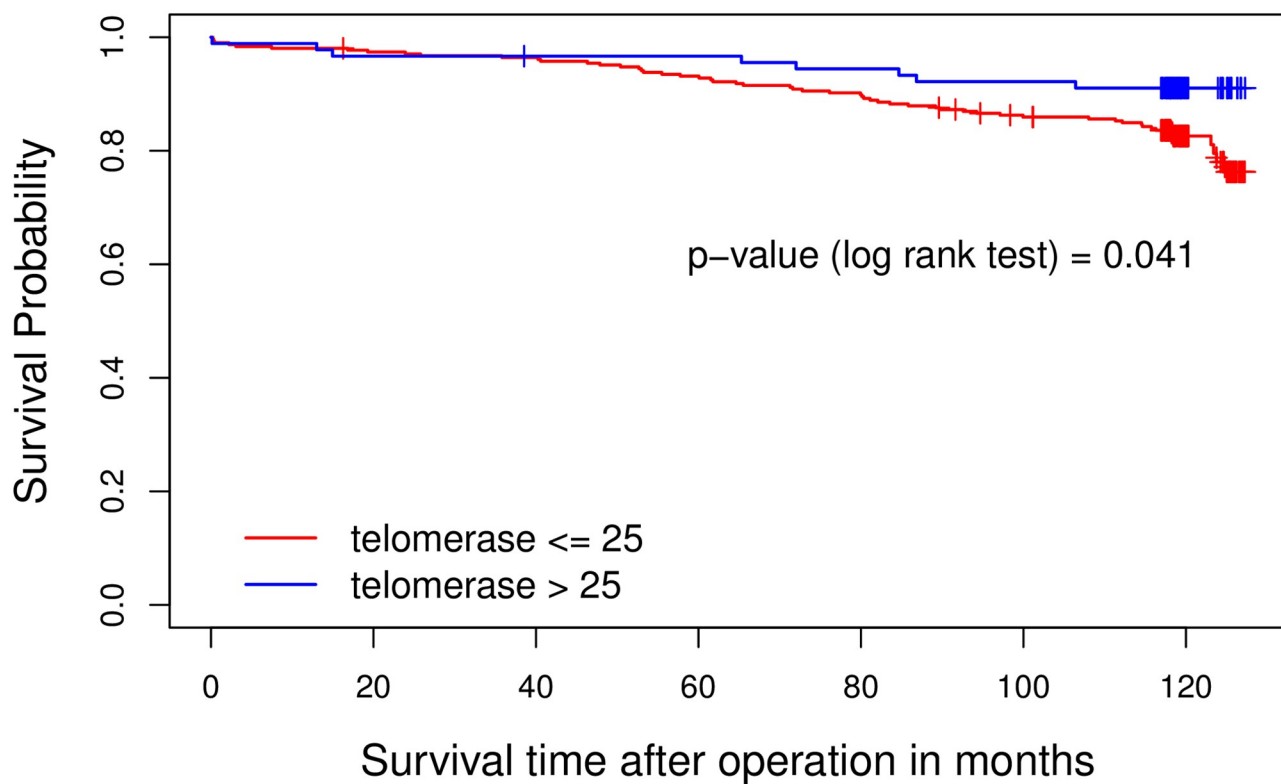

**Fig 1. The relationship between telomerase level and survival times in patients who underwent coronary artery bypass graft surgery.**

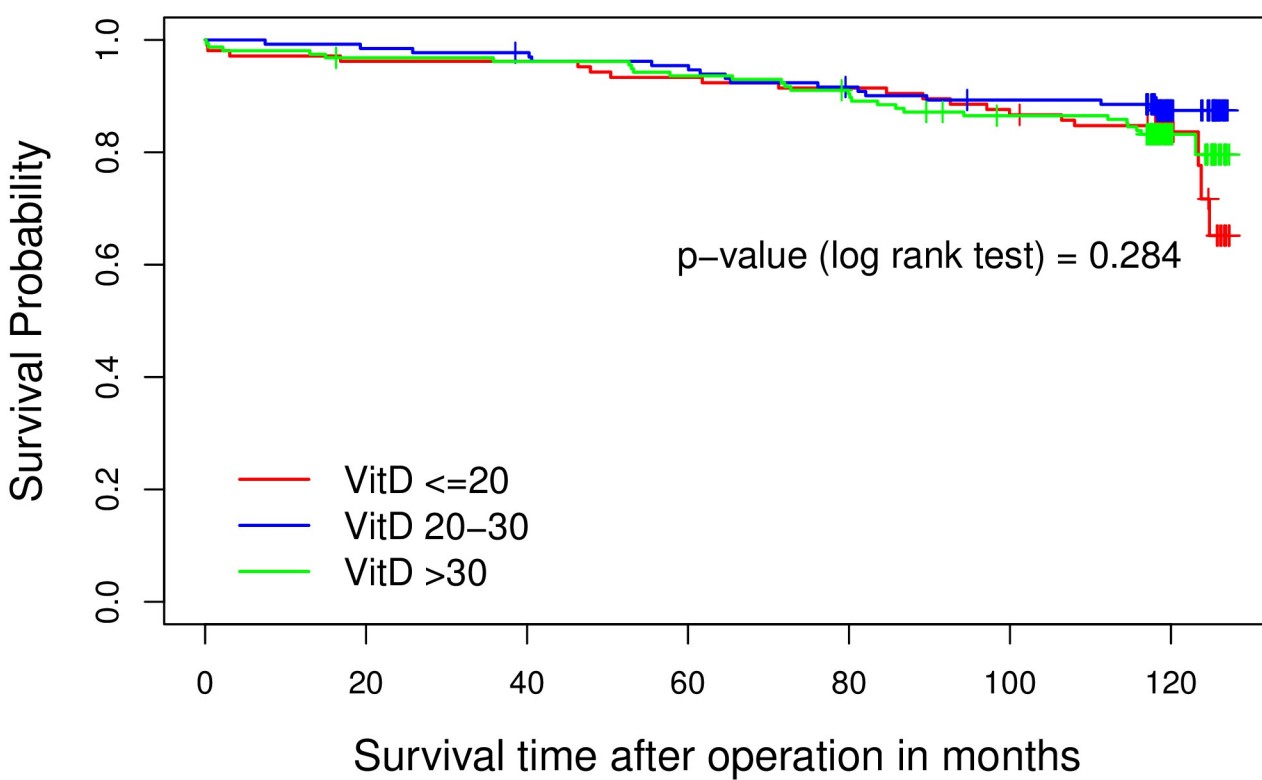

**Fig 2. The relationship between 25-hydroxy vitamin D level and survival times in patients who underwent coronary artery bypass graft surgery.**

outcome is in opposite direction[10]. Adaptation to long time deficiency of vitamin D in older patients and their higher chance of survival in the presence of low levels of vitamin D is another issue that has been reported[25]. This explanation fits well with our cohort; because it includes cardiac patients who are normally in their elderly.

## Strengths and limitations

Only three patients were lost to follow up for survival analysis which can be regarded as advantageous to this long-term study in which loss to follow up can be a major problem. We showed that serum level of telomerase was an independent predictor of mortality. It is promising as it can be easily used in clinical setting as a reliable measure for survival. Moreover, telomerase is probably more specific for evaluation of cardiac outcome compared to the telomere.

Our results showed lower telomerase may be associated with the history of MI, which is in accordance with the experimental findings and is reflected in our previous report as well[43]. We did not include emergent patients in our cohort. Therefore, further studies on patients with acute MI undergoing cardiac procedures is needed to verify if there is any relationship between the occurrence and extent of MI and telomerase level. We need to know if telomerase can be regarded as a determinant of myocardial protection alongside its role as a serum biomarker of outcome in cardiac patients.

We did only baseline measurement of 25(OH)D and telomerase. If we measured these values in time period between CABG and 10 years later, we could provide a better judgment about the effect of vitamin D and telomerase on the mortality. We assumed that the association of vitamin D insufficiency and mortality was hampered by patients' supplementation with vitamin D after cardiac surgery. However, it was probably too late to prevent from the effect of

**Table 3. Adjusted multivariate model for 10-year all-cause mortality using Cox regression.**

| Variable | level | Adjusted HR | 95% confidence interval | | P-value |
|---|---|---|---|---|---|
| | | | lower | upper | |
| *Demographic parameters* | | | | | |
| Gender | Woman | Ref | | | 0.504 |
| | Man | 1.304 | 0.598 | 2.840 | |
| Age (year) | | 1.070 | 0.033 | 0.109 | **0.000** |
| education | Illiterate/ Basic literacy | Ref | | | |
| | Elementary & high school | 0.708 | 0.340 | 1.475 | 0.357 |
| | Higher education | 0.544 | 0.231 | 1.279 | 0.163 |
| Diabetes | | 1.653 | 0.939 | 2.909 | 0.082 |
| PVD | | 1.789 | 1.037 | 3.086 | **0.036** |
| NYHA function class (II vs I) | | 0.922 | 0.598 | 1.422 | 0.713 |
| CVA | | 2.624 | 0.948 | 7.267 | 0.063 |
| Smoking | | 1.532 | 0.748 | 3.139 | 0.244 |
| Opium | | 3.092 | 1.022 | 9.358 | **0.046** |
| EF | | 0.963 | 0.939 | 0.988 | **0.003** |
| Creatinine | | 2.381 | 1.149 | 4.932 | **0.020** |
| Albumin | | 0.908 | 0.374 | 2.201 | 0.830 |
| CRP | | 1.049 | 1.008 | 1.092 | **0.019** |
| 25(OH)D | 20–30 | Ref | | | |
| | ≤20 | 1.124 | 0.534 | 2.365 | 0.759 |
| | >30 | 0.741 | 0.366 | 1.501 | 0.405 |
| Telomerase enzyme ≤25 effect differed by 25(OH)D | 20–30 | 0.093 | 0.010 | 0.871 | **0.037** |
| | ≤20 | 0.344 | 0.073 | 1.613 | 0.176 |
| | >30 | 1.146 | 0.413 | 3.178 | 0.793 |

EF, ejection fraction; CVA, cerebrovascular accident; NYHA, New York Heart Association; CRP, C-reactive protein, BUN, blood urea nitrogen; PVD, peripheral vascular disease; 25(OH)D, 25-hydroxy vitamin D

telomerase on the mortality. So, it is justified to correct any vitamin D insufficiency right before operation.

We also were concerned about long time of storage for serum samples in term of any decrease in the serum level of 25(OH)D and/ or telomerase. Some studies have addressed this and a study by Bodnar et al. showed that long-term storage over 40 years even if deteriorate 25 (OH)D somewhat, it did so similarly across samples[44]. We did not find the level of 25(OH) D in our samples to be lower compared to our fresh samples in our other studies at the same hospital [7,8,41]. Regardless, as the samples are at the same age and have been stored in the same conditions, the impact if any is expected to be the same for all and would not change their association of vitamin D/ telomerase with the outcome.

## Conclusion

The findings of this study showed that serum telomerase concentrations can be regarded as a potential predictor for long-term outcome in CAD patients who underwent CABG. Moreover, the association of telomerase with the mortality was significant at insufficient levels of 25(OH) D. Given the association between telomerase and MI, further studies are needed to clarify the significance of telomerase concentrations in short-term and long-term outcomes of patients

with recent MI. The results of such studies may help stratifying patients based on predicting their response to coronary artery disease.

## Supporting information

**S1 Dataset. Data set.**
(SAV)

## Acknowledgments

The authors would like to thank all patients who participated in this study.

## Author Contributions

**Conceptualization:** Mahdi Najafi, Mohamad Hassan Javanbakht, Arash Shirvani.

**Formal analysis:** Yun-Hee Choi, Mehdi Yaseri.

**Funding acquisition:** Mohamad Hassan Javanbakht.

**Investigation:** Mahtab Zarei, Elnaz Movahedi.

**Methodology:** Mahdi Najafi, Mohamad Hassan Javanbakht, Arash Shirvani.

**Supervision:** Mahdi Najafi, Mohamad Hassan Javanbakht, Frank W. Sellke, Saverio Stranges.

**Writing – original draft:** Mahtab Zarei, Mahdi Najafi, Yun-Hee Choi.

**Writing – review & editing:** Mahtab Zarei, Mahdi Najafi, Elnaz Movahedi, Mohamad Hassan Javanbakht, Yun-Hee Choi, Mehdi Yaseri, Arash Shirvani, Frank W. Sellke, Saverio Stranges.

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
