## [Decision Letter · Decision Letter 0]

13 Jul 2020

PONE-D-20-17025

The predictive role of circulating telomerase and vitamin D for long-term survival in patients undergoing coronary artery bypass grafting surgery (CABG)

PLOS ONE

Dear Dr. Najafi,

Thank you for submitting your manuscript to PLOS ONE. After careful consideration, we feel that it has merit but does not fully meet PLOS ONE’s publication criteria as it currently stands. Therefore, we invite you to submit a revised version of the manuscript that addresses the points raised during the review process.

We look forward to receiving your revised manuscript.

Kind regards,

Gaetano Santulli, MD

Academic Editor

PLOS ONE

Journal Requirements:

Reviewers' comments:

Reviewer's Responses to Questions

**Comments to the Author**

1. Is the manuscript technically sound, and do the data support the conclusions?

Reviewer #1: Partly

2. Has the statistical analysis been performed appropriately and rigorously? 

Reviewer #1: Yes

3. Have the authors made all data underlying the findings in their manuscript fully available?

Reviewer #1: Yes

4. Is the manuscript presented in an intelligible fashion and written in standard English?

Reviewer #1: Yes

5. Review Comments to the Author

Reviewer #1: Overall, the manuscript is well written and straight forward. I have just a couple of comments and suggestions.

1. The labels must have been mixed up for Men and Women population in Table 1.

2. Please elaborate on the process of determining the cutoff value for telomerase, i.e. 25 nM/L, using decision tree analysis. Is there biological or physiological rationale for this cutoff value?

3. Please include manufacturer information on the ELISA kits used.

4. Clarification on the units used for telomerase, i.e. nM/L. This must be a mistake. Molarity (M) is defined as the number of moles per L (mol/L). Telomerase is expressed as nmol/L, which is could also be written as nM, not nM/L.

5. Telomerase was found to have significant association with survival in the presence of insufficient levels of 25(OH)D (20-30 ng/mL). Why is the association lost when 25(OH)D is lower, i.e. <20 ng/mL?

6. The authors mention that serum telomerase can serve as an “earlier” predictor of cellular longevity than telomere length. How early is early for this to be considered a better alternative? Telomerase levels have been shown to gradually decrease with age, but increases significantly within an hour of exposure to stress. Thus, this means that the values of telomerase may fluctuate greatly, and will not be persistently high or low for a considerable period of time. Hence, there’s a huge possibility that the timing of the sample collection may significantly affect the result of a test.

7. The authors concluded that serum telomerase concentrations can be regarded as a predictor for long-term outcome in CAD patients who underwent CABG. However, considering that there were other variables (factors), e.g. age, opium addiction, history of CVA, PVD, serum creatinine, CRP, and cardiac EF, that were found significantly associated with the study outcome that may confound this observation, it would be difficult to ascertain if indeed serum telomerase is a good predictor for long-term outcomes in CAD patients. Hence, the authors should rephrase this statement and perhaps reiterate that it could be a “potential” predictor, and would warrant further investigation.

8. An inclusion criteria for this study is that subjects should have underwent CABG because they had at least 50% arterial obstruction based on their angiography in one or more vessels. If available, the authors should look into the size of arterial obstruction and the number of blocked vessels as potential confounders that may affect the study outcome.

6. PLOS authors have the option to publish the peer review history of their article (what does this mean?). If published, this will include your full peer review and any attached files.

Reviewer #1: No

---

## [Author Response · Author response to Decision Letter 0]

24 Jul 2020

Reviewer #1: Overall, the manuscript is well written and straightforward. I have just a couple of comments and suggestions.

Reply: Thank you for very kind remarks about our manuscript.

1. The labels must have been mixed up for Men and Women population in Table 1.

Reply: We rectified this issue in Table 1.

2. Please elaborate on the process of determining the cutoff value for telomerase, i.e. 25 nM/L, using decision tree analysis. Is there biological or physiological rationale for this cutoff value?

Reply: We didn’t find any cut-off values from the literature. There is no report on normal level of telomerase as well to our knowledge. Hence, we used the best statistical model for predicting outcomes that fits our data. Decision tree which is used in machine learning showed that two categories with one cut-off value might be the best model. We calculated cut-off point with ROC curve.

3. Please include manufacturer information on the ELISA kits used.

Reply: We added manufacturer information about the ELISA kits used in our study.

4. Clarification on the units used for telomerase, i.e. nM/L. This must be a mistake. Molarity (M) is defined as the number of moles per L (mol/L). Telomerase is expressed as nmol/L, which is could also be written as nM, not nM/L.

Reply: Thanks for your comment. We changed telomerase unit to nmol/L all over the text.

5. Telomerase was found to have significant association with survival in the presence of insufficient levels of 25(OH)D (20-30 ng/mL). Why is the association lost when 25(OH)D is lower, i.e. <20 ng/mL?

Reply: Further investigations on CAD patients with different levels of 25(OH)D and telomerase would be needed to answer this question. We already know that the relationship between vitamin D and outcome is not a linear one and is rather U-shaped. If other factors determine outcomes at lower levels of 25(OH)D is an issue yet to be clarified. Limited statistical power might also be responsible for the lack of significant associations at the lowest levels of vitamin D. We added an explanation pertaining to this in discussion.

6. The authors mention that serum telomerase can serve as an “earlier” predictor of cellular longevity than telomere length. How early is early for this to be considered a better alternative? Telomerase levels have been shown to gradually decrease with age, but increases significantly within an hour of exposure to stress. Thus, this means that the values of telomerase may fluctuate greatly, and will not be persistently high or low for a considerable period of time. Hence, there’s a huge possibility that the timing of the sample collection may significantly affect the result of a test.

Reply: We agree with this reviewer that fluctuations of telomerase matter. Single measurement may be insufficient to judge how these fluctuations work. As such, we have acknowledged this issue as an additional limitation of our study. However, the timing for taking blood samples was the same among the patients, they did not have any significant stress in preoperative period, and we exclude any emergency surgery from the study. 

Circulating serum telomerase changes undoubtedly faster than telomere length. To the best of our knowledge there is no study on relationship of telomerase and telomere changes in cardiovascular diseases. We know that psychological stress mainly affect activity (not levels) and long term stress is associated with shorter telomere length. Although there is ample evidence about the effect of oxidative stress on telomerase level and telomere length, our knowledge about the potential role of psychological stress is scarce. 

We already know about the damaging effect of stress on patient survival. The important point is how the stress can impact survival. Telomerase is probably one of the mechanisms. Short-term changes of telomerase does not negate a long-term effect. It is evident from similar situations like the effect of stress on blood glucose, insulin, cytokines, cortisol and other biomarkers. However, a better understanding of the telomerase role needs further investigations.

7. The authors concluded that serum telomerase concentrations can be regarded as a predictor for long-term outcome in CAD patients who underwent CABG. However, considering that there were other variables (factors), e.g. age, opium addiction, history of CVA, PVD, serum creatinine, CRP, and cardiac EF, that were found significantly associated with the study outcome that may confound this observation, it would be difficult to ascertain if indeed serum telomerase is a good predictor for long-term outcomes in CAD patients. Hence, the authors should rephrase this statement and perhaps reiterate that it could be a “potential” predictor, and would warrant further investigation.

Reply: Following this reviewer’s suggestion, we have rephrased the concluding paragraph accordingly. 

8. An inclusion criteria for this study is that subjects should have underwent CABG because they had at least 50% arterial obstruction based on their angiography in one or more vessels. If available, the authors should look into the size of arterial obstruction and the number of blocked vessels as potential confounders that may affect the study outcome.

Reply: We have already mentioned the number of diseased vessels even though it did not fulfill the requirement to be included in multivariate analysis. We did not add information pertaining to the size of obstruction and/or blocked vessels. We observed in our previous study on CABG patients that details of coronary artery involvement calculated by Gensini score did not provide additional information to our simpler categorization based on three vessel structures.

---

## [Editor Report · Decision Letter 1]

28 Jul 2020

The predictive role of circulating telomerase and vitamin D for long-term survival in patients undergoing coronary artery bypass grafting surgery (CABG)

PONE-D-20-17025R1

Dear Dr. Najafi,

We’re pleased to inform you that your manuscript has been judged scientifically suitable for publication and will be formally accepted for publication once it meets all outstanding technical requirements.

Kind regards,

Gaetano Santulli, MD

Academic Editor

PLOS ONE
---

## [Editor Report · Acceptance letter]

5 Aug 2020

PONE-D-20-17025R1 

The predictive role of circulating telomerase and vitamin D for long-term survival in patients undergoing coronary artery bypass grafting surgery (CABG) 

Dear Dr. Najafi:

I'm pleased to inform you that your manuscript has been deemed suitable for publication in PLOS ONE. Congratulations! Your manuscript is now with our production department. 

Kind regards, 

on behalf of

Dr. Gaetano Santulli 

Academic Editor

PLOS ONE